# Under-screened Aboriginal and Torres Strait Islander women's perspectives on cervical screening

Tamara L. Butler[1,2]*, Natasha Lee[1,2], Kate Anderson[1,2], Julia M. L. Brotherton[3,4], Joan Cunningham[2], John R. Condon[2], Gail Garvey[1,2], Allison Tong[5], Suzanne P. Moore[2,6], Clare M. Maher[7], Jacqueline K. Mein[8], Eloise F. Warren[9], Lisa J. Whop[2,10]

**1** Faculty of Medicine, School of Public Health, The University of Queensland, Brisbane, Australia, **2** Wellbeing and Preventable Chronic Diseases Division, Menzies School of Health Research, Charles Darwin University, Casuarina, Northern Territory, Australia, **3** Melbourne School of Population and Global Health, University of Melbourne, Melbourne, Victoria, Australia, **4** Australian Centre for the Prevention of Cervical Cancer, Carlton, Victoria, Australia, **5** Sydney School of Public Health, The University of Sydney, Sydney, New South Wales, Australia, **6** Molly Wardaguga Research Centre, College of Nursing and Midwifery, Charles Darwin University, Casuarina, Northern Territory, Australia, **7** Southern Queensland Centre of Excellence in Aboriginal and Torres Strait Islander Primary Health Care, Brisbane, Queensland, Australia, **8** Tropical Public Health Services Cairns, Cairns and Hinterland Hospital Health Service, Queensland Health, Cairns, Queensland, Australia, **9** Yerin Eleanor Duncan Aboriginal Health Centre, Wyong, New South Wales, Australia, **10** National Centre for Epidemiology and Population Health, Australian National University, Australian Capital Territory, Canberra, Australia

* t.butler@uq.edu.au

**Data Availability Statement:** Data cannot be made publicly available for ethical reasons. Participants did not consent to public release of full transcripts and the nature of the qualitative data are such that

## Abstract

Aboriginal and Torres Strait Islander women have lower participation in Australia's National Cervical Screening Program than other Australian women. Under-screened (including never screened) women's voices are rarely heard in research evidence, despite being a priority group for interventions to increase cervical screening participation. This study aimed to describe under-screened Aboriginal and Torres Strait Islander women's perspectives on cervical screening. Participants were 29 under-screened (women who had either never screened, had not screened in the previous five years or had recently screened in the past three months after more than five years) Aboriginal and Torres Strait Islander women from five communities across three states/territories. Female Aboriginal and Torres Strait Islander researchers Yarned with women about why they did not participate in screening and how to improve screening. Yarning is an Indigenous qualitative research method in which relationships and trust facilitate culturally safe conversation. Transcripts were analysed thematically. The proportion of eligible women who screened within 30 days after the Yarn was calculated. We identified four themes describing how the harms outweighed the benefits of cervical screening for under-screened women. These were: 1) distress, discomfort, and trauma; 2) lack of privacy and control; 3) complicated relationships with health care providers (HCPs); and 4) pressured, insensitive, and/or culturally unsafe communication from HCPs. Under-screened women who had recently screened had maintained privacy and control through self-collection and had experienced trauma-informed and empathetic care from their HCPs. While we cannot unequivocally attribute women's subsequent participation in screening to their involvement in this study, it is notable that one third of eligible

public availability would compromise participant confidentiality. Requests for specific excerpts relating to published verbatim quotes will be considered on a case-by-case basis. Requests for data may be sent to the corresponding author (t. butler@uq.edu.au) and to the Human Research Ethics Committee (HREC) of the Northern Territory Department of Health and Menzies School of Health Research (ethics@menzies.edu.au; reference number: 2017-2993). Please note that the authors will need to seek approval from the six HRECs listed in the Ethics Approval section before releasing data.

**Funding:** This study was supported by the National Health and Medical Research Council (NHMRC) funded Centre of Research Excellence (CRE) in Targeted Approaches To Improve Cancer Services for Aboriginal and Torres Strait Islander Australians (TACTICS; #1153027), the NHMRC-funded CRE in Discovering Indigenous Strategies to improve Cancer Outcomes Via Engagement, Research Translation and Training (DISCOVER-TT; #1041111), and the Cancer Council NSW Strategic Research Partnership to improve cancer control for Indigenous Australians (STREP Ca-CIndA; SRP 1301, with supplementary funding from Cancer Council WA). The views expressed in this publication are those of the authors and do not necessarily reflect the views of the funders. TB was supported by an Australian Research Council Discovery Australian Aboriginal and Torres Strait Islander Award (#IN190100050) funded by the Australian Government. LJW was funded by a NHMRC Early Career Fellowship (#1142035) and a NHRMC Investigator Grant (#2009380). JCu was funded by a NHMRC Research Fellowship (#1058244). GG was funded by a NHMRC Investigator Grant (#1176651). AT was supported by a NHMRC Career Development Fellowship (#1106716). NL, KA, JB, JCo, SM, CM, JM, and EW received no specific funding for this work. The funders had no role in study design, data collection and analysis, decision to publish or preparation of the manuscript.

**Competing interests:** The authors have read the journal's policy and have the following conflicts: The study is affiliated with Southern Queensland Centre of Excellence in Aboriginal and Torres Strait Islander Primary Health Care, Wuchopperen Health Service Pty Ltd and Yerin Eleanor Duncan Aboriginal Health Centre. These commercial affiliations provided support in the form of salaries for authors CM, JM, and EW, respectively. This does not alter our adherence to all the PLOS ONE policies on sharing data and materials.

under-screened women were screened within 30 days after the Yarn. Enhancing privacy, implementing trauma-informed approaches to care and sensitivity to the clinician-client relationship dynamics could enhance women's sense of comfort in, and control over, the screening procedure. The opportunity to Yarn about cervical screening and self-collection may address these issues and support progress toward cervical cancer elimination in Australia.

## Introduction

Aboriginal and Torres Strait Islander women bear a greater burden of cervical cancer than other Australian women, with higher incidence and mortality rates (19.9 vs 10 per 100,000 women; and 7.8 vs 2.2 per 100,000 women, respectively) [1]. These differences can largely be explained by differences in access to and participation in cervical screening. Aboriginal and Torres Strait Islander women are less likely to participate in cervical screening than non-Indigenous women, and the participation gap is not closing over time [2–5]. In Queensland, approximately 50% of Aboriginal and Torres Strait Islander women had participated in screening in the five years to 2011 compared to approximately 80% of other Australian women [4]. While comprehensive national data are lacking, this suggests that a large proportion of Aboriginal and Torres Strait Islander women are under-screened or never screened. Compared with women who regularly participate in cervical screening, women who are under-screened or who have no screening history have a higher risk of developing cervical cancer [6], and survival is lower for women whose cervical cancer was not diagnosed through screening [7].

Since Australia's National Cervical Screening Program (NCSP) was introduced in 1991, cervical cancer incidence and mortality among Australian women has halved [1]. The NCSP renewal in December 2017 saw the two-year Papanicolaou test replaced by a five-yearly human papillomavirus (HPV)-based Cervical Screening Test (CST) for women aged 25 to 69 years (with exit testing between ages 70 to 74), new clinical follow-up pathways, and the option of self-collection for under- or never screened women, along with a National Cervical Screening Register [8]. The success of the NCSP has heralded predictions that Australia will be the first country to eliminate cervical cancer as a public health problem [9]. However, Aboriginal and Torres Strait Islander women may not reach the goal of cervical cancer elimination due to persistent systemic barriers to participation in screening [10,11].

In general, the barriers to participating in cervical screening reported by women relate to emotional responses (embarrassment; discomfort; fear of procedure and results); knowledge; prior experiences (previous experience of sexual violence; negative cervical screening experience); health care provider (HCP) characteristics (such as gender of the health professional; lack of empathy); and logistics (lack of time; forgetting; transport to clinic; cost) [12–14]. In addition to these barriers, Aboriginal and Torres Strait Islander women experience barriers related to culturally unsafe health care, including racism; culturally inappropriate education, information and communication from HCPs; and a lack of community consultation and perceived lack of privacy and confidentiality within clinics [15–18]. These barriers exist within the context of ongoing impacts of colonisation, intergenerational trauma, and broader inequities in health and wellbeing for Aboriginal and Torres Strait Islander people [19,20].

Evidence about barriers to cervical screening often represent the views of women who regularly screen. Consequently, trials and interventions aiming to increase participation in cervical screening may not be informed by the views of women for whom they are most needed.

Available data support the idea that barriers to cervical screening differ between under-screened and screened women [21], suggesting it would be inappropriate to rely on screened women's views when devising strategies to support under-screened women's participation. A qualitative synthesis of barriers to cervical screening in countries with a cervical screening public health program found that just 4 of 39 qualitative articles focused on never- or under-screened women [12]. The views of Aboriginal and Torres Strait Islander women are also under-represented: a systematic review found no quantitative studies reporting Aboriginal and Torres Strait Islander women's barriers to screening [13] and a qualitative synthesis found just one study [12].

This study aimed to describe the views and experiences of Aboriginal and Torres Strait Islander women who were never screened, under-screened, or recently-screened after a long interval in relation to cervical screening, including the factors that influence their decisions not to participate in cervical screening.

## Methods

### Study approach

The *Screening Matters*: *Aboriginal and Torres Strait Islander women's attitudes and perspectives on participation in cervical screening* study (for brevity, *Screening Matters*) was initiated, led, and run by Aboriginal and Torres Strait Islander women (LJW, TB, GG). The study approach centred Aboriginal and Torres Strait Islander women's perspectives and captured individual, community, and structural influences on women's participation in cervical screening. Details of this approach are reported elsewhere [15,22,23] and adhere to COnsolidated criteria for REporting Qualitative research (COREQ) guidelines [24].

### Participants

Seventy-nine eligible Aboriginal and Torres Strait Islander women participated in the larger *Screening Matters* study. Information about participant recruitment via Primary Health Care Clinic (PHCC) staff and data collection is reported in detail elsewhere [15]. Data reported in this paper were from 29 Aboriginal and/or Torres Strait Islander women who self-reported in the pre-Yarn survey that they had either 1) never screened, 2) had not screened in the previous five years or 3) had recently screened in the past three months after an interval longer than five years. The latter group was included in the current analysis as they could provide insight into factors that influenced their decision to recommence screening. When necessary to distinguish between groups, we refer to this group as "recently screened women". For brevity, we respectfully refer to the participants collectively as "under-screened women".

### Data collection

After obtaining written informed consent, participants completed a short demographic and health survey. Next, an Indigenous qualitative research methodology called Yarning was used. Yarning is a conversational and relaxed style of sharing stories and information among Aboriginal and Torres Strait Islander people. It is characterised by both social and research-focused conversation. Researchers and participants share information about themselves to build trust and accountability in a culturally safe way and participants provide information in a flexible, narrative format [25,26]. Aboriginal and Torres Strait Islander women conducted all Yarns. A semi-structured Yarning guide provided topics and key questions but allowed researchers to explore topics as they arose. Key topics included personal and community knowledge and views about screening, reasons for not screening, decision-making processes,

experiences with the PHCC, and suggestions to improve screening. Women's perceptions of the self-collection test were also explored and these have been reported elsewhere [23]. Yarning guide questions relevant to the current analysis are provided in S1 File.

After the site visits had been completed, each PHCC key contact provided the de-identified number of under-screened or never screened women who screened in the 30 days following the Yarn. The denominator (n = 21) does not include four recently screened women, one woman who did not have a record at the PHCC, and three women whose PHCC did not respond to requests for data. The total proportion of women who had screened since participating in the Yarn was calculated. The method of participation (self-collected or clinician-collected) was not requested.

## Qualitative data analysis

Thematic analysis was conducted using NVivo [27] to organise data. Following initial guidance from an experienced qualitative researcher (KA), a comprehensive coding structure and codebook were developed. Aboriginal and Torres Strait Islander women conducted the analysis. TB and NL independently coded five transcripts, meeting after each to discuss, revise the code structure and update the codebook. TB and NL then independently coded all Yarns. Following this, the nodes were distributed and summarised independently between TB, NL, and LW. As analysis progressed, analysts met regularly to debrief and discuss concepts emerging from the data (using collaborative Yarning [25]). This allowed a culturally safe space to reflect, share and decompress from the stories shared by women, and to reflect on our positioning as Aboriginal and Torres Strait Islander researchers and the intersection of our relatedness and identity with the women's Yarns. During this process we also identified nodes which required all three researchers to discuss the interpretation and outcomes of resulting analysis. TB then synthesised overarching themes from the node summaries and reflective discussions with iterative feedback and reflection with all co-authors.

## Ethics approval

Ethics approval for this research was obtained from the Aboriginal Health and Medical Research Council of New South Wales (AH&MRC) Ethics Committee (1341/17), Central Australian Human Research Ethics Committee (CAHREC, CA-18-3113), Far North Queensland Human Research Ethics Committee (FNQ HREC, HREC/18/QCH/41-1218), Human Research Ethics Committee of the Northern Territory Department of Health and Menzies School of Health Research (2017–2993), Metro South Human Research Ethics Committee (MSHREC, HREC/18/QPAH/52) and the University of Queensland HREC (2021/HE002399). Participants provided informed written consent to participate in the research.

## Results

### Participant characteristics

All participants identified as women. Most women had not screened for more than five years (79.3%, n = 23), while some had never participated in cervical screening (6.9%, n = 2). A further 13.8% of women (n = 4) had recently screened in the preceding three months after an interval longer than 5 years. Ages ranged from 26 to 66 years old with a median age of 49.5 years old. Most women identified as Aboriginal (89.7%, n = 26), had children (79.3%, n = 23), completed year 12 schooling or below (58.6%, n = 17), and spoke English at home (75.9%, n = 22). Two-fifths (41.4%, n = 12) were single; about a third (31.0%, n = 9) were in a de facto relationship or married; 13.8% (n = 4) were separated or divorced; 10.3% (n = 3) were

widowed and 1 woman (3.4%) declined to answer this question. Women mostly resided in major cities (65.5%, n = 19); 31% (n = 9) resided in regional or remote areas. Women were from Queensland (51.7%, n = 15), New South Wales (31%, n = 9) and the Northern Territory (17.2%, n = 5), across five participating PHCCs.

Of the 21 women for whom cervical screening participation information was provided, 7 (33%) women had participated in screening in the 30 days following the Yarn. While women's subsequent participation in screening cannot be directly attributed to their involvement in this study, it is notable that during the research Yarns, some women suggested that the chance to discuss screening can support a decision to participate.

> *Well, this* [the research] *has sort of pushed me to get a Pap smear, like it's made me want to go and get it done, talking about it more now.* P81

Thematic analysis of the Yarns (illustrated in Fig 1) revealed that the harms of cervical screening outweighed the benefits for under-screened women. The harms were: 1) triggering distress, discomfort and trauma; 2) undermined privacy and control; 3) complicated relationships with HCPs; and 4) pressured, insensitive and culturally unsafe communication with HCPs. Amongst these themes, there were stories of women who had recently participated in screening because the barriers to screening were addressed.

## Harms outweigh benefits of cervical screening

Women reported advantages of screening including the potential early detection and prevention of cancer, peace of mind about health, being a role model for female family members and

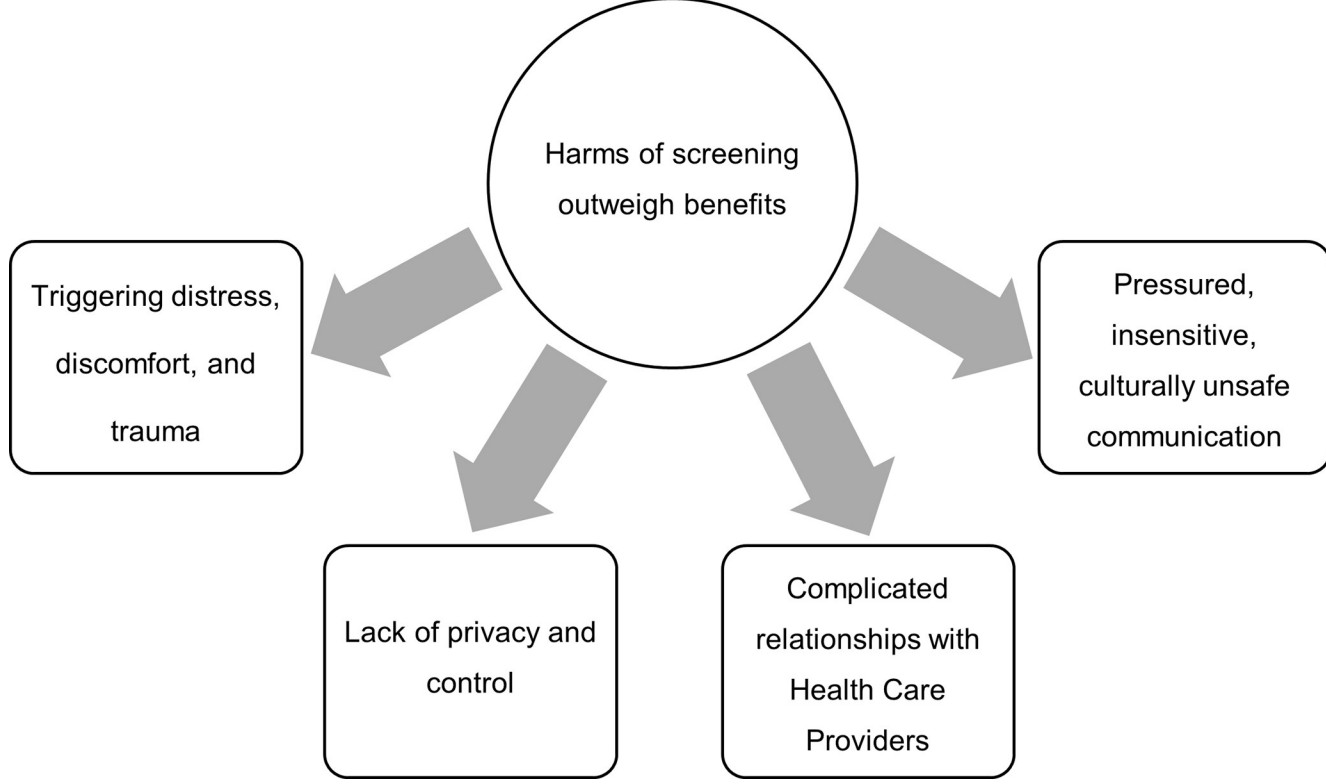

**Fig 1. Key themes underlying under-screened women's views of cervical screening.**

breaking a cycle of poor health. The benefits of screening were perceived to enhance the health of future generations. Many women felt that they knew enough or did not want or need further information about screening. However, for many women, knowledge and awareness of the advantages of cervical screening was not enough to outweigh the harms they associated with screening.

For most there was a perception that participating in screening would compromise their sense of control over psychological and physical boundaries. Women held their understandings of the benefits and their reasons for not participating in cervical screening simultaneously in mind, and actively acknowledged the conflict this created. Ultimately this tension manifested in women's decisions not to participate in screening.

> *I find it's a bit intrusive. It's embarrassing. I feel shame. But I know on the other hand, too, from an intellectual perspective I know it's important that we do get these things done so that we look after ourselves as people. P14*

The following qualitative analysis describes the barriers to cervical screening underlying women's decisions not to participate. The stories of women who had recently participated in screening are also described to illustrate how some barriers were addressed.

## Triggering distress, discomfort, and trauma

**Heightened worry.**   Women generally spoke about the process of cervical screening, including the test itself, in ways that indicated heightened distress, fear, and worry. Women's descriptions of cervical screening were characterised by feelings of violation, invasiveness, discomfort, pain, and intimidation that stemmed from their physical and psychological experiences and perceptions of the test. Women felt that they must be physically and mentally prepared for feelings of discomfort, exposure, and vulnerability during cervical screening. Opportunistic screening was sometimes declined due to not feeling "*prepared for that today*" (P50). Women talked about needing to "*psych up*" (P05) and "*build up*" (P30) courage for cervical screening. Women expressed feelings of shame about the intrusive and embarrassing nature of cervical screening and feelings of vulnerability it provoked.

> *You really have to psych yourself up to go in and go on that table, take your bottom half [. . .] clothes off and then to be spread eagle, it's quite embarrassing and then to have somebody to be touching you in those areas and that I just find is very intrusive as a person. P14*

Women felt distressed about several aspects of the cervical screening process. For some, panic set in with the arrival of reminder letters, while others felt frightened and anxious thinking about cervical screening. One woman reported that her fear about screening was exacerbated by the drawn-out process of organising and attending the appointment for the clinic, emphasising that getting it over and done with quickly would prevent her ruminating and second-guessing her decision to screen. Women felt fear thinking about the potential results and follow-up appointments, which one woman described as "*horrifying*" (P53). For some, the strength of these emotions was increased by experiences of trauma. As well as distress, women experienced physical discomfort caused by the cervical screening test itself.

**Negative past experiences.**   For some women, past and/or first experiences of cervical screening were traumatic, and these experiences were formative in their decisions not to participate in screening. Women emphasised how daunting the cervical screening test and speculum seemed during the first experience as a young woman, with feelings of fear being heightened due to not knowing what to expect. Women described experiences when the

cervical screening test and the reasons for it were not explained, leading to feelings of shame, embarrassment, and awkwardness. Women were usually young and were screened opportunistically at the time of these events. Some women's past negative experiences of screening were compounded by physical and psychological effects of sexual violence.

> *I guess the first one* [cervical screening] *that I had when I was 19, [. . .] I can still remember the day as clear as day, it was very unexpected. I went in for a pregnancy test [. . .] He basically just said, "Well, while you're here we'll do this." Bang. Done. And I walked out thinking, oh, my God. What have I just done? And with my history* [of sexual violence] *it was just so bad. Then it was like, that's it. I'm never trusting another doctor again. I'm never getting on that bed without asking what they're going to do first.* P02

**Association with sexual violence among some women.** A small number of women in this sample explained that experience of sexual violence impacted on their decisions to screen. For these women, cervical screening triggered a distress response.

> *P*: *For me, it just triggered trauma.* [Cervical screening] *triggered that horrible hurt feeling internally. Like it was just—- -*
>
> *Q: So it sort of brought up those past experiences?*
>
> *P*: *Yes, both physical and emotional, yes.* P50

Women described avoiding cervical screening to guard their privacy and as a measure of protection against reliving psychological and physical trauma.

One under-screened woman who worked as a HCP at a clinic expressed a need for HCPs to be more aware of how cervical screening may trigger trauma among women who have experienced sexual violence.

> *There needs to be a certain sort of training for the doctors in that space particularly those who have experienced sexual trauma, just being aware of what those triggers might be. Because I do work in that space. Experiences of working with survivors. Something like that can be very triggering. We don't know what they've been through.* P46

## Lack of privacy and control

Cervical screening was viewed as personal and private business. Lack of privacy (including in the clinic environment, the intimate nature of the test itself or with their personal information) was a major factor in women's decision to not participate in cervical screening. Women described needing to be reassured of privacy and security through having rooms with locking doors, not just a curtained-off area, and saw benefits in a women's only clinic. Women described that discretion was also an important part of ensuring privacy, preferring a "*quiet word*" (P74) about cervical screening, rather than a HCP publicly "*singing out for me*" (P74) in a way the community would know about. While this was particularly emphasised by women in remote areas, some women in urban areas also preferred the community not to know the purpose of their visit to the clinic. One woman wanted to understand where her sample and information went, who looks at it, and how she can get a copy to stay informed of her results and also ensure confidentiality.

Women emphasised the importance of their independent decision making, autonomy, and agency regarding screening and its implications for feeling a sense of control over their bodies.

*I'm just an independent woman and make up my own mind. If I want to do it, I'll do it. It's as simple as that*. P68

One woman's recent decision to screen was directly related to feelings of control and privacy the option of self-collection afforded her. She viewed clinician-collected screening as an uncomfortably intimate procedure, in part, due to her exposure to sexual violence. She had not participated in cervical screening for over 30 years. However, when her doctor facilitated the option for self-collection for her, she completed it without hesitation.

*It was a private thing and it was, you know, the doctor wasn't doing it. So, I felt more comfortable, because I did it, you know? [. . .] Because there's no one there, you know what I mean? Looking at you*. P19

## Complicated relationships with health care providers

**Trusting relationships.** Women described their relationships with HCPs as a complex factor in the decision-making process relating to screening. Trusting relationships were very important to women, and many had searched hard to find a HCP they could trust. They valued long-lasting relationships with HCPs because this allowed time for trust to be built and saved frustrations involved in repeating medical history to new doctors. Trust was fundamental to even broaching the subject of cervical screening.

*No, because it* [discussion of cervical screening] *hasn't come up because I haven't found a reliable doctor or someone that I really trust. You've got to find that. It takes time for that.* P26

However, at times, women's trusting relationships with HCPs added complexity to their decisions to participate in cervical screening. Some felt having cervical screening done by their usual HCP could jeopardise their valued and trusting relationships with HCPs because the process was too intimate and embarrassing for both the HCP and the woman.

One woman's recent decision to screen was influenced by her need to preserve the strong relationship she had with her doctor after many years of "doctor shopping". "Doctor shopping" was triggered by previous doctors raising the subject of cervical screening, and it allowed her to avoid disclosing experiences of sexual violence which surfaced with discussions of screening.

*I guess for a very long time to be honest, I doctor shopped if they started raising the subject with me, it was, oh, time to find a new doctor. Because I didn't want to have to tell them all the time that this [childhood sexual violence] is why I don't have them done. So I just would jump from doctor to doctor then when I [. . .] started seeing the doctor regularly and [. . .] built that trust, I was then able to tell her, and she had a really good understanding, so I stopped doctor shopping.* P02

To preserve the trusting relationship they had built over time and respect the woman's wishes, the HCP referred her to another provider for cervical screening, after sudden changes to the woman's period prompted clinical investigation.

*My thing was that I couldn't get my doctor to do it and then still come back and look at her and have that same relationship, [. . .] we were well talked up about that and I certainly*

*indicated to her straight out that I couldn't come back to her if she was the one to do it. And that was just because of the relationship that I have with her.* P02

**Women's business.** Women generally viewed cervical screening as Women's Business (referring to distinct roles, customs, and cultural practices reserved for and shared amongst women only [28]). They preferred a female HCP to discuss and conduct screening, citing reasons including cultural appropriateness, female HCPs having the same body parts, and feelings of safety and understanding.

*If it's a female doctor I don't have a problem. [. . .] I'm uncomfortable with a male doing it to be honest. I don't know why but, you know, we all look the same pretty much, do you know what I mean. It makes no difference to them, but it makes a difference to me. Just a personal thing, personal choice. I would rather a female always.* P54.

Women reported a need for greater availability of female HCPs to conduct screening, and a lack of availability of female HCPs as a barrier to cervical screening. Many reported that they would not attend a screening appointment with a male HCP. One woman described her experience of booking the appointment in the hope that a female HCPs would be available.

*I walked* [into the clinic for cervical screening appointment] *and it was a male doctor and he's like, "Why are you here?" and I'm like, "I don't know why I'm here", and then I walked out, so I think it was—and I was just praying like please be a female, please be a female and it's so hard to get booked in with a female doctor, so I think that's why it makes it a bit harder as well.* P30

Women spoke of their past experiences of screening conducted by male HCPs. One woman took a "*get it over and done with*" (P68) attitude to screening with a male HCP, while another spoke of a situation in which a male HCP declined to perform cervical screening.

*Yeah that's the worst part, you have to go talk to usually a male, and then you see, "Oh, we don't do them here," it's like they make it a big thing because you're a female asking a male and they go like, "I don't do that here, and you need to go somewhere else to do that."* P30

Other women reported that their male HCP never broached the subject of cervical screening with them at all but were persistent about other health issues.

*I know myself that I should be getting it done but if my doctor was on my case as well, the fact that he hasn't even mentioned it in all of the years and he's on my case about my diabetes and my weight loss and everything else but yeah, whether he just doesn't feel comfortable doing them with his patients, I don't know, but if he had have been on my case I probably would have had it done.* P43

Some women's objections to the cervical screening process itself was so strong that the gender of the HCP was irrelevant. Even with a female HCP, the shame associated with cervical screening was a barrier to participating in screening.

## Pressured, insensitive, and culturally unsafe communication with HCPs

Excessive and overt pressure from HCPs to participate in cervical screening was not welcomed and undermined women's sense of decision-making power and control. Pressure to screen

was particularly not welcome from male HCPs. Women used phrases such as "*people forcing you*" (P74) and being "*harped on*" (P12) to describe this sense of pressure. Women felt that HCPs sometimes pressured them with little regard for concurrent health issues that women were managing.

> *Sometimes you're talked into having the test done. Like at the moment I'm dealing with mental health issues and so I really don't feel comfortable dealing with those things as well as having to have a Pap smear done because staff members are wanting this to be done at this timeframe sort of thing and it feels a bit like bullying.* P14.

Pressure sometimes caused women to push back and decline screening more strongly than they might have otherwise, with negative consequences for clinic attendance.

> *I've got a good relationship [*with clinic staff*], but I don't like to be harped on. You harp, I won't come back. I never came back here for a good 12 months because I got sick and tired of, "*[name]*, you have to have this.* [name], *you have to have that." Harp and I won't come back.* P12

Some reported a complete absence of communication about cervical screening from their HCPs. For others, it had been mentioned with HCPs but not discussed with the desired depth or empathy.

> *I notice when people say you need a Pap smear [. . .] they're like, You need to go get it done", that's pretty much really it. "You have to go and get it done, it needs to be done." It's not gentle.* P30

Racism from HCPs, experienced personally or by peers, made women feel unsafe. One woman reported an experience of racism during follow-up of abnormal Pap smear results where she felt the gynaecologist asked her questions about whether she had experienced sexual violence purely because she identified as Aboriginal. Because of this, the woman then refused to discuss the issue with the gynaecologist any further.

> *She asked me really invasive questions. [. . .] She asked if I'd been sexually abused. [. . .] Like I'm going to tell her. [. . .] And I'm thinking, how could you even—you don't even know me. [. . .] I was floored. [. . .] I usually don't have an issue talking about that but when she said it like that I was like, "No, I haven't." [. . .] But don't you dare ask me.* P53

Women expressed a desire for clear, specific and empathetic communication from HCPs about the purpose of cervical screening, and what to expect and do during the test, such as body positioning. There was a need for simple, friendly and accessible language without medical jargon when discussing cervical screening.

> *A friendly person* [would make me feel comfortable]. *Not the one that talks the rough way and too many hard words. When you're Aboriginal, you need someone that talks. Yeah, need more words that, you know, we can understand. A lot of people, they use too many hard words.* P74

Sensitive HCP communication and empathetic, trauma-informed care was central to one woman's recent positive experience of screening after a 24-year interval following experience

of sexual violence. This woman felt that this experience of care would make future cervical screening tests much easier.

[The nurse] *was just so in tune. She actually explained to me, "Okay, so when the apparatus hits that part of your cervix, it's going to send all sorts of messages to the brain. It's not going to know what to do with it." I was, "Is that what's happening." [...] She gave me some breathing techniques, she actually told me where [the speculum] was positioned. [...] and I took my partner with me, and she didn't care. I'm like, "My partner is coming in." "Yes, that's fine."* P50

## Discussion

We Yarned with 29 Aboriginal and/or Torres Strait Islander women from five PHCC settings in three States/Territories who were under-screened. Our findings are grounded in the lived experience of many under-screened Aboriginal and Torres Strait Islander women, and they provide insight into the complexity of decision making about screening. We found that whilst many under-screened women were aware of the advantages of participating in cervical screening, many felt that these were outweighed by the psychological and emotional harms they associated with screening. These harms included that cervical screening could trigger distress, discomfort and trauma; a lack of privacy and control; complicated relationships with HCPs; and pressured, insensitive, and culturally unsafe communication from HCPs. One third of eligible women participated in screening in the 30 days following the Yarn, suggesting that Yarning about screening may support women's decisions to screen. These findings directly inform how providers and systems need to change to provide culturally safe screening services supportive of women's needs.

### A need for trauma-informed care

Women's expressions of heightened distress and fear about cervical screening indicate that trauma-informed approaches to cervical screening service delivery may support women's decisions to screen. A trauma-informed approach involves recognising the far-reaching impacts of trauma, recognising signs of trauma and responding appropriately, and a focus on avoiding re-traumatization in health care settings [29]. Key tenets include safety; trustworthiness and transparency; peer support; collaboration and mutuality; empowerment, voice and choice; and consideration of cultural, historical and gender issues [29]. In the current study, many recently screened women's stories were characterised by trauma-informed approaches (e.g., trusting relationships, HCPs acknowledging women's emotions, co-creating solutions, ensuring choice and control). This empathetic approach to care may address many under-screened women's negative emotional and psychological associations with screening. This approach may also include practical measures to reduce distress, such as allowing extra time for appointments to provide space for sensitive conversations to occur and providing clear explanations of the procedure both before and during the event. Indeed, the process of Yarning with researchers about screening may have provided the time and space for the seven (33%) under-screened women to deeply consider the procedure and decide to participate in screening. A trauma-informed approach to care would also address women's reported concerns about pressured, insensitive and culturally unsafe communication from HCPs about cervical screening.

It is important to note that while some women reported sexual violence as a traumatic factor influencing their decisions not to participate in cervical screening, this does not describe the experience of all women who are underscreened. Clinicians should not presume

experience of sexual violence among Aboriginal and Torres Strait Islander women, nor that such experience is a reason that women may not participate in cervical screening, while also being cognisant that for others, experience of sexual violence could indeed be the reason for non-participation. Clinicians should maintain an open and non-judgemental approach to facilitating screening.

## Relationship with health care providers

Women's relationships with HCPs had both positive and negative impacts on screening. Women's trust in their HCP was fundamental in starting a discussion about cervical screening at all, but for some women, the intimate cervical screening procedure risked jeopardising their relationship with their HCP. Aboriginal and Torres Strait Islander women who screen regularly [15] and HCPs [22] also note the complexities of the clinician-client relationship in the context of cervical screening. Strong and trusting relationships with HCPs have been found to be critical in other women's health areas such as antenatal and postpartum care [30,31]. Accommodating the client's preference for an alternative unfamiliar HCP to conduct screening, including availability of a female HCP, is also an issue raised in other populations [12–14,32]. Understanding the importance of building strong and trusting relationships with health care providers, while recognising the complexities of not wanting to jeopardise the relationship with an intimate procedure may help HCPs facilitate safe and comfortable cervical screening for under-screened Aboriginal and Torres Strait Islander women.

## Privacy and control

Women expressed a need to maintain privacy and control over their screening experience. The option of self-collection may provide under-screened Aboriginal and Torres Strait Islander women with the privacy and comfort they need to participate. However, despite self-collection being made available for under-screened women with NCSP renewal, 2018–2019 data suggests that fewer than 6000 self-collected tests have been processed [33]. Findings from the larger *Screening Matters* study suggest that many Aboriginal and Torres Strait Islander women were not aware of the option for self-collection [23]. Self-collection is acceptable to and feasible for Aboriginal and Torres Strait Islander women [23,34–36]; one participant in the current study had recently screened after a 30-year interval through self-collection. Furthermore other under-screened Australian women report a sense of empowerment through the ability to collect their own sample [37]. Together these findings indicate an urgent need for clinician-client discussions about the option for self-collection in order to support women's need for privacy and control over their screening experience, a factor that has encouraged other under-screened Australian women to screen [37]. Further, from mid-2022, the restrictions on the eligibility for self-collection will be removed, allowing this choice for all potential screening participants [38]. How providers frame this choice (as equivalent in accuracy rather than as an inferior test and as easy to collect) will be critical to its adoption by women who otherwise will remain under- or never screened. Education, communication and implementation strategies directly informed by the needs of the communities that could benefit most are urgently needed.

## Screening participation follow-up

The combined quantitative and qualitative data suggest that the opportunity to Yarn about cervical screening may support women's decisions to participate in screening. While Yarning was not necessarily the reason for subsequent screening participation, it is striking that one third of eligible women completed cervical screening within 30 days of taking part in the *Screening*

*Matters* study. The importance of Yarning about screening was also reflected in the qualitative data: some women commented that the opportunity to Yarn about screening helped them to reflect on their reasons for declining screening and motivated them to organise screening. For some, the *Screening Matters* research project was the first opportunity they had had to Yarn about cervical screening at length and it had given them cause to reflect on participation in cervical screening. The finding suggests that one of the key mechanisms HCPs could use to support under-screened women is to engage them in open and non-judgemental discussions about their reasons for declining screening and co-create solutions together, as demonstrated by some recently screened women's stories. This could be supported by longer appointment times, ensuring workforce availability, and availability of women's spaces in which to discuss cervical screening. A shortage of appropriate workforce to provide more appointments and longer appointment times has previously been noted as a barrier to screening [22].

## Limitations

As recruitment focused on women attending PHCCs, the views and experiences of under-screened women not engaged with a PHCC are not reflected in the findings; these women may have different concerns or experiences about participation in cervical screening (e.g., access to the clinic or perceptions of clinical care). PHCCs that were involved in the study may also have an interest in supporting women's screening, so the sample may not represent the views of women in PHCCs where screening is not viewed as a priority. The increased attention on cervical screening in preparation for the researchers' visit may have also caused a surge in PHCC staff promotion of cervical screening which may not represent usual practice. Another limitation is that women self-reported their screening status and were often unable to recall exactly how long ago their previous cervical screen was, if they had had one. Because of this we are unable to report exactly how long ago (beyond five years) women had last screened.

## Strengths

A key strength of the *Screening Matters* study is the central positioning of Aboriginal and Torres Strait Islander women's voices as leaders, researchers, and participants in the study, bringing cultural understanding and sensitivity to the findings. Furthermore, the study was made possible through partnership with five Aboriginal Community Controlled Health Organisations and staff. The large number of under-screened Aboriginal and Torres Strait Islander women's voices reflected in this study is another key strength. Under-screened women's voices are under-represented in the literature [12,13]. The current findings provide strategies to support their participation in cervical screening that are guided by their views and preferences.

## Conclusions

Whilst Australia is on a trajectory to be one of the first countries in the world to eliminate cervical cancer as a public health problem (incidence< 4 per 100,000 women) [9,39,40], cervical screening remains critically low for Aboriginal and Torres Strait Islander women [4]. Reducing barriers to cervical screening is vital to achieve the 67% reduction in incidence required to reach elimination for Aboriginal and Torres Strait Islander women) [11,40]. This study provides important insights to develop strategies to engage this group, based on our findings that enhancing privacy, implementing trauma-informed approaches to care, ensuring sensitivity to clinician-client relationship dynamics and culturally safe communication styles could enhance women's sense of comfort in, and control over, the screening procedure. The option for self-collection in the renewed NCSP could facilitate positive and empowering screening experiences for under-screened women. Implementing these strategies will support under-screened

Aboriginal and Torres Strait Islander women to participate in screening and contribute to achieving cervical cancer elimination among all Australian women.

## Supporting information

**S1 File. Interview guide.**
(DOCX)

## Acknowledgments

We wish to thank the participants involved in this research study. Ownership of Aboriginal and Torres Strait Islander knowledge and cultural heritage is retained by the informant.

## Author Contributions

**Conceptualization:** Tamara L. Butler, Kate Anderson, Julia M. L. Brotherton, Joan Cunningham, John R. Condon, Gail Garvey, Allison Tong, Suzanne P. Moore, Lisa J. Whop.

**Formal analysis:** Tamara L. Butler, Natasha Lee, Kate Anderson, Lisa J. Whop.

**Funding acquisition:** Kate Anderson, Julia M. L. Brotherton, Joan Cunningham, John R. Condon, Gail Garvey, Allison Tong, Suzanne P. Moore, Lisa J. Whop.

**Investigation:** Tamara L. Butler, Lisa J. Whop.

**Methodology:** Tamara L. Butler, Kate Anderson, Julia M. L. Brotherton, Joan Cunningham, John R. Condon, Gail Garvey, Allison Tong, Suzanne P. Moore, Lisa J. Whop.

**Project administration:** Tamara L. Butler, Clare M. Maher, Jacqueline K. Mein, Eloise F. Warren, Lisa J. Whop.

**Resources:** Clare M. Maher, Jacqueline K. Mein, Eloise F. Warren.

**Supervision:** Lisa J. Whop.

**Visualization:** Tamara L. Butler.

**Writing – original draft:** Tamara L. Butler, Lisa J. Whop.

**Writing – review & editing:** Tamara L. Butler, Natasha Lee, Kate Anderson, Julia M. L. Brotherton, Joan Cunningham, John R. Condon, Gail Garvey, Allison Tong, Suzanne P. Moore, Clare M. Maher, Jacqueline K. Mein, Eloise F. Warren, Lisa J. Whop.

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
