## [Decision Letter · Decision Letter 0]

6 Jul 2022

Under-screened Aboriginal and Torres Strait Islander women’s perspectives on cervical screening

PONE-D-22-05427

Dear Dr. Butler,

We’re pleased to inform you that your manuscript has been judged scientifically suitable for publication and will be formally accepted for publication once it meets all outstanding technical requirements. Congratulations on your work!

Within one week, you’ll receive an e-mail detailing the required amendments. There were several grammar or style changes identified by one reviewer, but this manuscript overall is exceptional. When these have been addressed, you’ll receive a formal acceptance letter and your manuscript will be scheduled for publication.

Kind regards,

Andrea Knittel

Academic Editor

PLOS ONE

Reviewers' comments:

Reviewer's Responses to Questions

**Comments to the Author**

1. Is the manuscript technically sound, and do the data support the conclusions?

Reviewer #1: Yes

Reviewer #2: Yes

2. Has the statistical analysis been performed appropriately and rigorously? 

Reviewer #1: Yes

Reviewer #2: Yes

3. Have the authors made all data underlying the findings in their manuscript fully available?

Reviewer #1: Yes

Reviewer #2: Yes

4. Is the manuscript presented in an intelligible fashion and written in standard English?

Reviewer #1: Yes

Reviewer #2: Yes

5. Review Comments to the Author

Reviewer #1: The article is clearly written and the findings explained with adequate quotes from participants to highlight the points made.

As a Māori woman it is great to see the article lead author is an Aboriginal scholar and for clear Indigenous methodologies frame the research.

I have no required amendments.

I look forward to further work in this area.

Reviewer #2: From my reading, I believe this research and manuscript to be of a very high standard. The methods were thorough and the research design was well justified and fitting for the research question being addressed. The paper flowed and was easy to read, yet thorough and very informative. Congratulations.

6. PLOS authors have the option to publish the peer review history of their article (what does this mean?). If published, this will include your full peer review and any attached files.

Reviewer #1: No

Reviewer #2: **Yes: **Mrs Tegan Dutton

---

## [Editor Report · Acceptance letter]

22 Aug 2022

PONE-D-22-05427 

Under-screened Aboriginal and Torres Strait Islander women’s perspectives on cervical screening 

Dear Dr. Butler:

I'm pleased to inform you that your manuscript has been deemed suitable for publication in PLOS ONE. Congratulations! Your manuscript is now with our production department. 

Kind regards, 

on behalf of

Dr. Andrea Knittel 

Academic Editor

PLOS ONE